# The Transgene Expression of the Immature Form of the HCV Core Protein (C191) and the LncRNA MEG3 Increases Apoptosis in HepG2 Cells

**Dina Mofed** [1,2], **Salwa Sabet** [3], **Ahmed A. Baiomy** [3] and **Tamer Z. Salem** [1,4,5,*]

[1] Molecular Biology and Virology Lab, Biomedical Sciences Program, Zewail City of Science and Technology, October Gardens, 6th of October City, Giza 12578, Egypt
[2] Zoology Graduate Program, Department of Zoology, Faculty of Science, Cairo University, Giza 12613, Egypt
[3] Department of Zoology, Faculty of Science, Cairo University, Giza 12613, Egypt
[4] Department of Microbial Genetics, Agricultural Genetic Engineering Research Institute (AGERl), Agricultural Research Center (ARC), Giza 12619, Egypt
[5] National Biotechnology Network of Expertise (NBNE), Academy of Science Research and Technology (ASRT), Cairo 11334, Egypt
[*] Correspondence: tsalem@zewailcity.edu.eg; Tel.: +20-1014114122

**Abstract:** Long non-coding RNAs (lncRNAs) are regulated in cancer cells, including lncRNA MEG3, which is downregulated in Hepatocellular Carcinoma (HCC). In addition, hepatitis C virus (HCV) core proteins are known to dysregulate important cellular pathways that are linked to HCC development. In this study, we were interested in evaluating the overexpression of lncRNA MEG3, either alone or in combination with two forms of HCV core protein (C173 and C191) in HepG2 cells. Cell viability was assessed by MTT assay. Transcripts' levels of key genes known to be regulated in HCC, such as *p53*, *DNMT1*, miRNA152, *TGF-b*, and *BCL-2,* were measured by qRT-PCR. Protein expression levels of caspase-3 and MKI67 were determined by immunocytochemistry and apoptosis assays. The co-expression of lncRNA MEG3 and C191 resulted in a marked increase and accumulation of dead cells and a reduction in cell viability. In addition, a marked increase in the expression of tumor suppressor genes (*p53* and miRNA152), as well as a marked decrease in the expression of oncogenes (*DNMT1*, BCL2, and *TGF-b*), were detected. Moreover, apoptosis assay results revealed a significant increase in total apoptosis (early and late). Finally, immunocytochemistry results detected a significant increase in apoptotic marker caspase-3 and a decrease in tumor marker MKI67. In this study, transgene expression of C191 and lncRNA MEG3 showed induction in apoptosis in HepG2 cells greater than the expression of each one alone. These results suggest potential anticancer characteristics.

**Keywords:** LncRNA MEG3; HCV core protein (C191); HCV core protein (C173); HepG2 cells; transfection; apoptosis

## 1. Introduction

Hepatocellular carcinoma (HCC) is the fourth cause of cancer-related death worldwide. It was reported that more than one million people may die from HCC by 2030, according to the World Health Organization [1]. Infections with the B or C hepatitis virus (HBV or HCV), chronic alcohol consumption, and non-alcoholic fatty liver disease (NAFLD) are the major risk etiological factors of HCC [2].

On the other hand, HCV is a global health issue with about 71.1 million chronically infected people, accounting for 1% of the population worldwide. Hepatitis C virus is an enveloped virus with a positive-strand RNA genome of approximately 9.6 kb that encodes a long polyprotein of approximately 3000 amino acids (aa) [3]. Core protein (C) is among the structural proteins of HCV and was found to have multi-functional activities in HCC development and progression. The core protein interacts with proto-oncogenes and alters their expression, resulting in hepatocarcinogenesis [4,5].

Full-length HCV core 191 (C191), an immature form of HCV core, localizes in the cytosolic side of ER [6–9]. The processing and maturation of HCV core occur in the ER and depend mainly on the interaction with the ER membrane. The maturation of the HCV core depends on the cleavage of the signal peptide at the site of 173aa through the signal peptidase in the ER lumen resulting in the production of a mature form of HCV core protein (C173) [10,11], which localizes in the nucleus [12,13]. A previous study showed that the accumulation and processing of immature HCV core (C191) in the ER cause cell apoptosis by inducing ER stress, which leads to the induction of the apoptotic pathway via overexpression of C/EBP homologous protein (CHOP). CHOP overexpression causes induction of apoptosis and translocation of Bax to mitochondria. Bax activation causes mitochondrial membrane depolarization, cytochrome c release, caspase-3 activation, and apoptosis triggering [6,14].

On the contrary, the localization of truncated mature form of HCV core (C173) in the nucleus may induce HCC proliferation by different pathways, including the downregulation of tumor suppressor miRNA152, which targets oncoprotein wnt-1 [15], suppression of tumor suppressor P16 by upregulating DNA methyltransferase 1 (DNMT1), and DNMT3b that increases the hyper-methylation of P16's promoter [16]. Moreover, HCV core protein induces HCC progression by promoting the upregulation of transforming growth factor-β1 (TGF-β1) transcription, where it binds at bases $-376$ to $-331$ bp on the promoter region of TGF-β1 and activates its expression [17]. It is worth noting that the C173 protein could inhibit apoptosis by triggering the expression of BCL-2 by different pathways, such as activating the STAT3 signaling cascade [18].

Long non-coding RNAs (lncRNAs) have been implicated in the development and progression of HCC. LncRNAs are transcripts with over 200 bases lacking protein-coding capacity [19]. Various HCC-related lncRNAs were found to exhibit abnormal expression and play a role in malignant phenotypes such as persistent proliferation, evasion of apoptosis, accelerated vascular formation, and gain of invasive potential [20]. Various lncRNAs such as NEAT1, MALAT1, ANRIL, HULC, HOTAIR, MVIH, and SNHG1 were upregulated in HCC samples. On the contrary, several tumor suppressor lncRNAs, namely CASS2 and MEG3, are downregulated in HCC [21]. On malignant hepatocytes, a microarray study of >23,000 lncRNAs was conducted, 713 (3%) of which were downregulated, including the maternally expressed gene 3 (MEG3), which its expression was decreased by 210-fold compared to expression in non-cancerous hepatocytes [22]. Other previous studies showed that lncRNA MEG3 suppresses HCC in vitro and in vivo. For example, real-time quantitative PCR results showed that lncRNA MEG3 levels were much lower in CCl4-induced mice liver fibrosis models and human fibrotic livers [23]. Moreover, overexpression of lncRNA MEG3 decreases cell viability and increases apoptosis in HCC cell lines via targeting MDK/miR-9-5p and regulation of the AKT/PDK pathway [24]. The LncRNA MEG3 expression was shown to be altered during viral infection; for example, in non-small cell lung cancer (NSCLC) tissues, the increased level of expression of lncRNA MEG3 inhibits the expression level of TLR4, subsequently reducing RSV (respiratory syncytial virus) infection-induced inflammatory responses [25]. In addition, lncRNA MEG3 can influence the progression of SARS-CoV-2 through ADAR protein (adenosine deaminase acting on RNA). ADAR plays an important role in innate immunity against viral infection, and lncRNA MEG3 acts as the main regulator of ADAR [26]. There is a relationship between HCV core protein and lncRNA MEG3, as the mature form of HCV core protein (C173) inhibits the expression of lncRNA MEG3 by different pathways. The C173 protein downregulates tumor suppressor miRNA 152 [15], and this triggers the expression of DNMT1 [27]. DNMT1 has a vital role in suppressing lncRNA MEG3 via methylation of the MEG3 promoter, resulting in increasing HCC proliferation and progression [28]. On the contrary, the immature form of HCV core (C191) shares with lncRNA MEG3 the induction of apoptosis and inhibition of cell growth via ER stress by increasing the expression of ER stress-related proteins such as GRP78, ATF6, PERK, IRE1, CHOP and cleaved-caspase-3 [6,14,29].

As the lncRNA MEG3 is downregulated during HCC, in this study, we were interested in understanding how each of the core forms (C173 and C191) in the presence of a high level of lncRNA MEG affects or regulates apoptosis and other HCC key genes. This will provide better insights and understanding of the link between HCV infection and cancer development.

## 2. Materials and Methods

### 2.1. Plasmid Constructs and Cloning

The pCI-MEG3 plasmid was obtained from Addgene (Addgene plasmid # 44727), and pCI-MEG3 lacking MEG3 (pCIΔMEG3) was performed by deleting the MEG3 gene from the pCI-MEG3 vector by restriction enzymes EcoR1-HF and Not1 (NEB, Ipswich, MA, USA). pCIΔMEG3 vector of ~3964 bp size was purified from gel using MinElute gel extraction kit (Qiagen, Hilden, Germany), and then the gap was filled using Phusion high fidelity Taq polymerase (NEB, Ipswich, MA, USA) [30]. The product was ligated by T4 ligase (NEB, Ipswich, MA, USA) and transformed into E.coli DH5α cells.

C191 and C173 were subcloned in the pEGFP-N1 vector as a vehicle vector to transfect mammalian HepG2 cells. C191 and C173 sequences were amplified from a synthetic fragment of 700 bp that includes the full core gene and part of the E1 gene sequence. The synthetic fragment was manufactured according to the sequence of hepatitis C virus type 4a (acc. # Y11604.1) and by GenScript, Hong Kong. Before the insertion of C191 and C173 in the pEGFP-N1 vector, they subcloned in other vectors to facilitate their insertion in the pEGFP-N1 vector. The sequence corresponding to C191 was amplified by PCR using the primers 5′GCGAATTCAGCACGAATCCTAAACCT3′, and 5′TAGGTGGCCGAAGCGGGGACAAGT3′ and the following program; 95 °C for 30 s followed by 30 cycles of (95 °C for 30 s, 60 °C for 30 s, 72 °C for 2 min) and a final extension step at 72 °C for 5 min [30]. Then the PCR product was then cloned into a pDrive cloning vector by TA cloning using a PCR cloning kit (Qiagen, Hilden, Germany) before being transformed into competent E. coli DH5a. The sequence corresponding to C173 was amplified by PCR using the following primers 5′GCGAATTCAGCACGAATCCTAAACCT3′, which contains EcoR1-HF restriction sequence in bold, and 5′TGCTCTAGATTAGGAGCAACCGGGGAGATT3′, which contains Xba1 restriction sequence in bold. In the following program; 95 °C for 30 s was followed by 30 cycles (95 °C for 30 s, 55 °C for 30 s, 72 °C for 2 min) and a final extension step at 72 °C for 5 min [30]. The PCR product was also digested with restriction endonucleases EcoR1-HF and Xba1 (NEB, Ipswich, MA, USA). Moreover, corresponding digestion was applied to pFastBac™ Dual Expression Vector (Invitrogen Life Technologies, Carlsbad, CA, USA). The C191 sequence was subcloned into pEGFP-N1 (Clontech, Palo Alto, CA, USA) between KPN1 and HindIII endonucleases (NEB, Ipswich, MA, USA), and the C173 sequence was subcloned into p EGFP-N1 between EcoR1-HF and pst1 (NEB, Ipswich, MA, USA), and corresponding digestion was applied to plasmid pEGFP-N1 (Clontech, Palo Alto, CA, USA). Digested products were isolated and then ligated with T4 DNA ligase (NEB, Ipswich, MA, USA) to obtain pN1-C191 and pN1-C173 plasmids. The ligated products were transformed into competent *E. coli* DH5a cells. Nucleotide sequencing of the cloned gene was validated by Eurofins Genomics, Germany.

### 2.2. Cell Culture and Transfection

The HCC cell line HepG2 was obtained from Nawah scientific research, Cairo, Egypt. Cells were cultured in RPMI medium, supplemented with 10% FBS (Gibco, Thermo Fisher Scientific, Waltham, MA, USA), and incubated at 37 °C with 5% $CO_2$. Cells were seeded at a density of 150,000–200,000 cells per well in 24-well plates for transfection with vectors pCI-MEG3, pCIΔMEG3, pEGFP-N1, pN1-C173, pN1-C191, and co-transfected with pCI-MEG3+pN1-C191 and pCIMEG3+pN1-C173. Lipofectamine 3000 transfection reagent was used for the transfection (Invitrogen, Thermo Fisher Scientific, Waltham, MA, USA), and 0.5 µg DNA for each recombinant vector was used. For co-transfection, 0.25 g DNA of each recombinant vector was used. Before adding the transfection complexes to the cells, they

were prepared in an opti-mem reduced serum medium (Gibco, Thermo Fisher Scientific, Waltham, MA, USA) before incubation at RT for 30 min [30]. The transfected cells were cultured for 48–72 h at 37 °C with 5% $CO_2$.

### 2.3. Examination of Cell Morphology

Leica DMi8 inverted microscope (Leica Microsystems, Wetzlar, Germany) was used to examine cell morphology for 72 h after transfection with the vectors mentioned above with a magnification of 20 X, with an excitation wavelength of 488 nm and an emission wavelength of 509 nm. Photos were taken by Leica Application Suite X (LAS X) software [30].

### 2.4. Cytotoxicity Analysis and Cell Viability by MTT Assay

Cell viability and toxicity of transfected HepG2 cells with the different vectors mentioned above were estimated using a3-(4,5-dimethylthiazol-2-yl)-2,5-diphenyltetrazolium bromide (MTT) assay, as described previously [31,32]. Cells were seeded at a density of 7000–10,000 cells per well in 96-well plates in RPMI 1640 medium, supplemented with 10% FBS for 24 h. The medium of each well was removed, and then cells were washed twice using 1.0 M phosphate-buffered saline (PBS; pH7.4) at 25 °C before transfection with each vector at a gradual concentration (0.05–0.3 μg) for individual transfections or co-transfections. Cells were incubated at 37 °C in a 5% $CO_2$ incubator for 48–72 h before adding 10 μL MTT solution (5 mg/1 mL of 1.0 M PBS pH 7.4), then cells were incubated for 4 h at 37 °C in a 5% $CO_2$ incubator. Media were discarded and replaced with 100 μL of dimethyl sulphoxide and incubated in a dark room for 2 h. The absorbance was measured at 570 nm with a plate reader (FLUOstar® Omega, BMG Labtech, Ortenberg, Germany). The percentage viability and the half-maximal inhibitory concentration (IC50) were calculated using GraphPad Prism 8 software (GraphPad Software, San Diego, CA, USA).

### 2.5. Apoptosis Assay

Apoptotic cell populations were measured by Annexin V-FITC apoptosis detection kit (Abcam, Cambridge, UK). After transfection of HepG2 cells with the vectors in 24-well plates by Lipofectamine 3000 transfection reagent as mentioned above in Section 2.2, the transfected HepG2 cells and control untransfected HepG2 cells were collected after 72 h by trypsinization and washed twice with PBS (pH 7.4). Cells were incubated in the dark with 0.5 mL of Annexin V-FITC/PI solution for 30 min in the dark at RT according to the manufacturer's protocol. Then cells were analyzed by ACEA Novocyte™ flow cytometer (ACEA Biosciences, San Diego, CA, USA) with FITC and PI fluorescent signals using FL1 and FL2 signal detector, respectively (λex/em 488/530 nm for FITC and λex/em 535/617 nm for PI). For each sample, 12,000 events were used, and positive FITC and/or PI cells were quantified by quadrant analysis and calculated by ACEA NovoExpress™ software (ACEA Biosciences, San Diego, CA, USA) [33].

### 2.6. Immunocytochemistry for Caspase-3 and MKI67

The protein expression of caspase-3 and MKI67 in transfected HepG2 cells with vectors mentioned above and control untransfected HepG2 cells was estimated by Immunocytochemical assay according to the manufacturer's protocol (Novus Biologicals, Littleton, CO, USA) with slight modifications. Briefly, after transfection of HepG2 cells with the vectors, which were mentioned above in 24-well plates by Lipofectamine 3000 transfection reagent as mentioned above in Section 2.2., the transfected media were removed, and HepG2 cells were washed twice with PBS (4 °C). Cells were fixed with 4% paraformaldehyde for 20 min, then washed three times with PBS. HepG2 cells were permeabilized with 0.3% Triton®-X (diluted in D-PBS with calcium and magnesium) for 5 min at RT. HepG2 cells were incubated overnight at 4 °C with anti-KI67 rabbit monoclonal MM1 at concentration 1:100 (Novocastra Laboratories Ltd., Newcastle, UK), and rabbit polyclonal antibody against cleaved caspase3 (Asp175), at concentration 1:50 (Cell Signaling Technology, Beverly, MA, USA) [34]. HepG2 cells were washed three times with PBS and incubated with horseradish

peroxidase (HRP)-conjugated goat secondary antibodies at 25 °C for 1 h, then washed by PBS three times each for 5 min while shaking.

Finally, HepG2 cells were stained with a DAB solution (3 ng/mL) for 10 min. Caspase-3 and MKI67 expression levels were examined by fluorescent microscopy using Leica DMi8 inverted microscope (Leica Microsystems, Wetzlar, Germany) at a magnification of 20×, with an excitation wavelength of 488 nm and an emission wavelength of 509 nm. Photos were taken by Leica Application Suite X (LAS X) software [30]. Quantification of Caspase-3 and MIK67 was estimated by measuring the percentage area of expression from three randomly chosen fields in each section using image analysis software (Image J, version 1.46a, Bethesda, MD, USA) [35].

### 2.7. Total RNA Extraction, cDNA Synthesis, and Quantitative Real-Time PCR (qRT-PCR)

Gene expression of key cellular genes that is related to apoptosis, proliferation, etc., including *MEG3, P53, DNMT1, miRNA152, TGF-B, BCL2, HCV core 191, HCV core 173*, and *β-Actin* genes, was assessed using qRT-PCR with primers presented in Table 1. The RNeasy Mini Kit (Qiagen, Hilden, Germany) was used to extract total RNA from transfected HepG2 cells with the vectors in 24-well plates by Lipofectamine 3000 transfection reagent as mentioned above in Section 2.2. NanoDrop™ was used to evaluate the concentration and purity of the isolated RNA (Thermo Fisher Scientific, Waltham, MA, USA). The RNA was treated on-column with DNase I RNase-free enzyme (Qiagen, Hilden, Germany) according to the kit's procedure during the extraction. Random primers were used to make the cDNA and then reverse-transcribed using a High-Capacity cDNA Reverse Transcription kit (Thermo Fisher Scientific, Waltham, MA, USA). Per reaction, we used 2 µg of isolated RNA. Quantitative real-time PCR was performed using PowerUP SYBR green master mix (Thermo Fisher Scientific, Waltham, MA, USA) and analyzed by a Quantstudio 12K Flex (Thermo Fisher Scientific, Waltham, MA, USA). The used program was as follows: 50 °C for 2 min, 95 °C for 2 min, and 40 cycles (95 °C for 1 is and 60 °C for 30 s) [30]. The forward primer of *MEG3* was designed based on the pCI-MEG3 vector (Addgene 44727) upstream of the *MEG3* gene and the reverse primer located inside the MEG3 gene itself. All the primers were designed to match most of the transcript variants of the tested genes.

**Table 1.** qRT-PCR primers of genes commonly regulated by MEG3 and HCV core.

| Gene | Forward Primer | Reverse Primer |
|---|---|---|
| *MEG3* | 5′CCACTCCCAGTTCAATTACAGCTC3′ | 5′TAGTGCCCTCGTGAGGTGTAG3′ |
| *P53* | 5′CAGCACATGACGGAGGTTGT3′ | 5′TCATCCAAATACTCCACACG3′ |
| *DNMT1* | 5′CCATCAGGCATTCTACCA3′ | 5′CGTTCTCCTTGTCTTCTCT3′ |
| *miRNA152* | 5′CCCAGGTTCTGTGATACACTCC3′ | 5′CTTCCGGGCCCAAGTTCTG3′ |
| *TGF-B* | 5′AAGAAGTCACCCGCGTGCTA3′ | 5′TGTGTGATGTCTTTGGTTTTGTCA3′ |
| *BCL-2* | 5′GGTGGGGTCATGTGTGTGG3′ | 5′CGGTTCAGGTACTCAGTCATCC3′ |
| *HCVcore 191 or 173* | 5′GATACATCCCGCTCGTAGGC3′ | 5′TTGATCCCGTCCTCCAAAGC3′ |
| *β-Actin* | 5′CATGGAGTCCTGTGGCATC3′ | 5′CAGGGCAGTGATCTCCTTCT3′ |

### 2.8. Statistical Analyses

At least three independent replicas for each assay were performed, and the average was calculated. One-way analysis of variance (ANOVA) and Tukey's multiple-comparison post-test was used to evaluate the significance of gene expression. Differences between groups were significant at a *p*-value of <0.05 and a *p*-value < 0.01. Statistical analyses were performed with GraphPad Prism 8.0 (GraphPad Software, San Diego, CA, USA).

## 3. Results

### 3.1. Overexpression of lncRNA MEG3 and Expression of C191 Reduce the Viability of HepG2 Cells

The cytotoxic effect of overexpression of lncRNA MEG3 and expression of C191 was evaluated in HepG2 cells at 48 h and 72 h post-transfection (pt) using MTT assay. A

significant decrease in cell viability was detected when cells were transfected with pCI-MEG3, pN1-C191, or both. The reduction efficiencies were as follows: (a) transfection with pCI-MEG3 by 34.2% and 44.2% at 48 h and 72 h (IC50 equivalent to 95.5 ng and 74 ng), respectively; (b) transfection with pN1-C191 by 30.5% and 34.7% at 48 h and 72 h (IC50 equivalent to 110 ng, and 93.6 ng), respectively; (c) co-transfection with pCI-MEG3 + pN1-C191 by 37.4% and 44.7% at 48 h and 72 h (IC50 equivalent to 99.8 ng and 93.1 ng), respectively. On the other hand, no significant effect was detected on cell viability post-transfection with pN1-C173. However, the cell viability was significantly decreased when pN1-C173 was co-transfected with pCI-MEG3 by 17.5% and 23% (IC50 equivalent to 130.6 ng and 125.8 ng) at 48 h and 72 h, respectively (Figure 1A). It is worth noting that according to the above results, the effect of lncRNA MEG3 varied considerably from 48 to 72 h, while the variation was not as vast for C191. In addition, the co-transfection with pCI-MEG3 + pN1-C191 did not show a considerable reduction percentage from transfection with only pCI-MEG3.

The results of cell morphology examination by inverted microscope showed signs of cell shrinkage, apoptotic bodies, and accumulation of dead cells after transfection of HepG2 cells with pCI-MEG3 or pN1-C191; these signs increased after co-transfection with pCI-MEG3 + pN1-C191 (Figure 1B).

### 3.2. Co-Expression of lncRNA MEG3 and C191 Induces Apoptosis in HepG2 Cells

Flow cytometry was used to determine the fraction of cells undergoing apoptosis (Figure 2). The results showed an increase in early and late apoptosis by 5.2- and 3.2-fold, respectively, post-co-transfection of HepG2 cells with pCI-MEG3+ pN1-C191 compared to the control (untransfected HepG2 cells) (Figure 2A). This increase in early and late apoptosis was maintained when compared to HepG2 cells transfected with pEGFP-N1 or pCIΔMEG3 by approximately 4.0- and 2.5-fold, respectively. Additionally, there was an increase in early and late apoptosis after transfection with pCI-MEG3 by 3.2- and 3.0-fold, respectively, compared to untransfected HepG2 cells and by 2.5- and 2.3-fold, respectively, compared to cells transfected with pCIΔMEG3.

A)

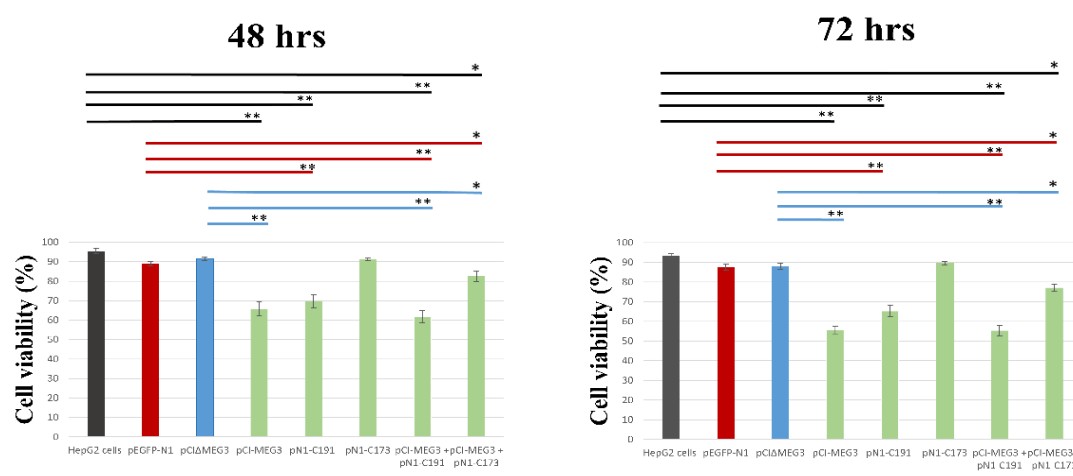

**Figure 1.** *Cont.*

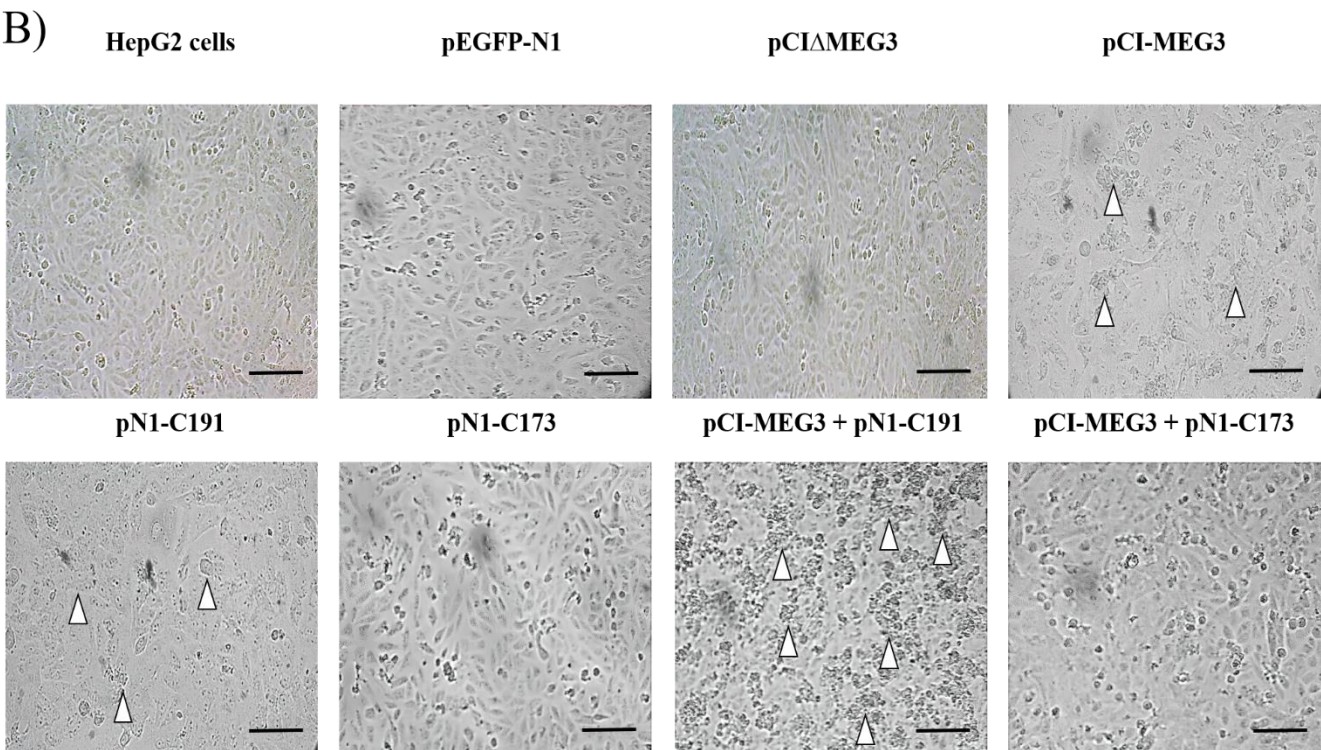

**Figure 1.** (**A**): Cell viability of transfected HepG2 cells with plasmids pEGFP-N1, pCIΔMEG3, p-CI-MEG3, pN1-C191, pN1-C173, p-CI-MEG3 + pN1-C191 and p-CI-MEG3 + pN1-C173 compared to control HepG2 cells, following incubation period 48 h and 72 h. The mean of three independent replicates was used to calculate these data values. Data presented as mean ± SD. (*) represents *p*-value < 0.05 and (**) represents *p*-value < 0.01 compared to control untransfected HepG2 cells, pEGFP-N1 and pCIΔMEG3. Colored lines represent the significance of cell viability in HepG2 cells between each sample and its control. Controls are divided into (a) HepG2 cells (Black), (b) HepG2 cells transfected with pEGFP-N1 (Red), and (c) HepG2 cells transfected with pCIΔMEG3 (Blue). (**B**): Representative Photomicrographs showing the viability of HepG2 cells transfected with different plasmids after an incubation period of 72 h with the magnification of 20×. (Δ) represents the accumulation of dead cells and apoptotic bodies. Scale bar 100 μm.

pN1-C191 showed an increase in early apoptosis by 3.5-fold, but no increase in late apoptosis was detected.

Co-transfection with pCI-MEG3 + pN1-C173 increased early and late apoptosis by 2.6- and 2.4-fold, respectively, compared to control untransfected HepG2 cells, and by approximately 2- and 1.7-fold, respectively, compared to pEGFP-N1 or pCIΔMEG3. Transfection with pN1-C173 did not cause an effect on early or late apoptosis (Figure 2A,B).

### 3.3. Co-Expression of lncRNA MEG3 and C191 Increases Caspase-3 and Decreases KI67 in HepG2 Cells

Immunocytochemical analysis of caspase-3 expression revealed a significant increase in expression of caspase-3 post-co-transfection with pCI-MEG3 + pN1-C191 compared to controls. The results showed that transfection with pCI-MEG3 or pN1-C191 significantly increased the expression of caspase-3 compared to controls. There was no marked effect post-transfection with pN1-C173. However, co-transfection with pCI-MEG3 + pN1-C173 significantly increased the expression of caspase-3, which showed that the presence of LncRNA MEG3 can mitigate the effect of HCV core 173 on the proliferation of HepG2 cells, as shown in (Figure 3A,B).

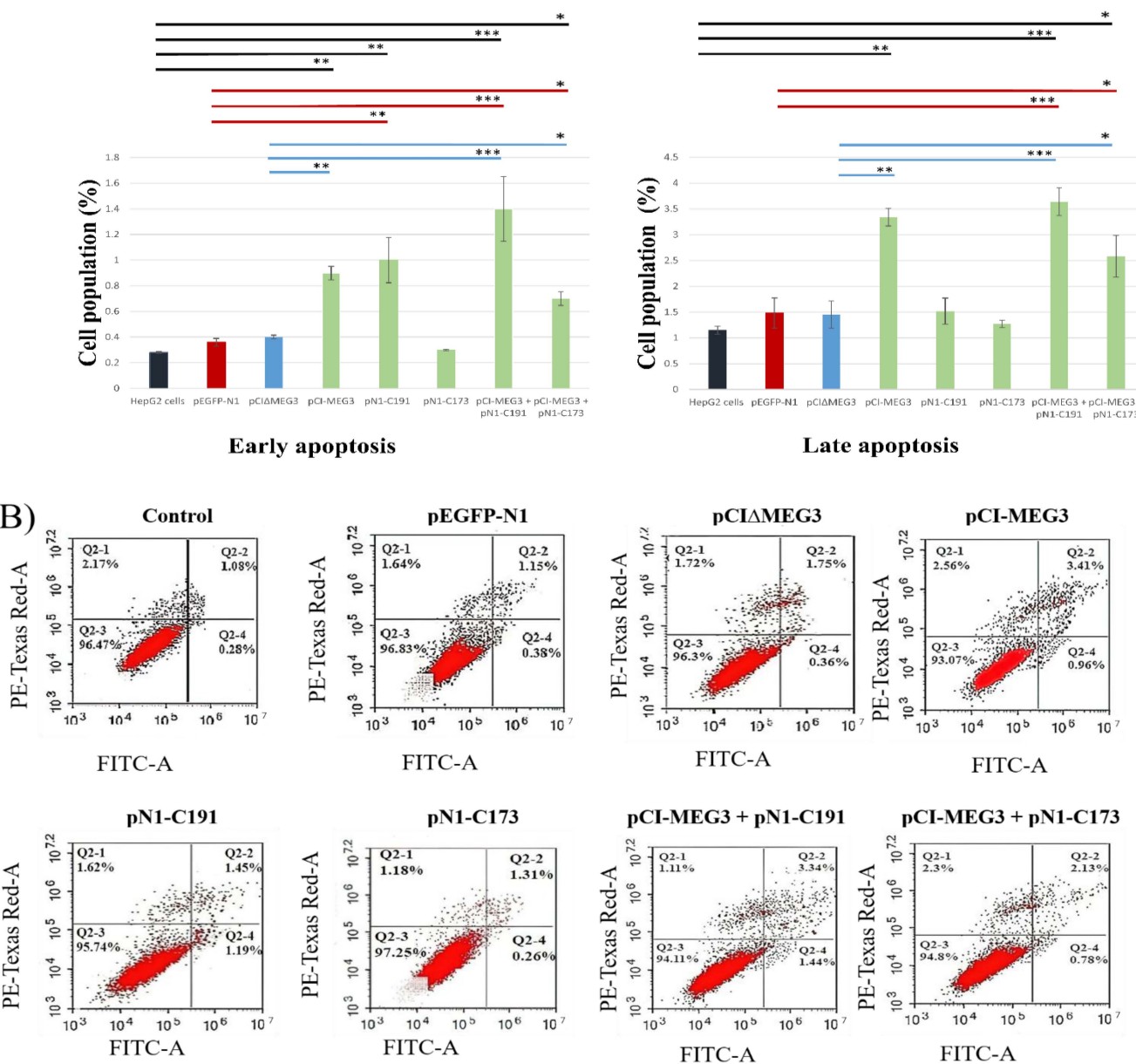

**Figure 2.** (**A**) Cell death mechanism in HepG2 after transfection with different vectors. Cells were exposed to transfection for 72 h. Percentages of cells undergoing apoptosis/necrosis were assessed using Annexin-V/FITC-PI staining compared to control cells. The chart of flow cytometry results is divided into four gates; Q2-1 represented necrosis, Q2-2 represented late apoptosis, Q2-3 represented intact normal cells, and Q2-4 represented early apoptosis, as shown in the figure. Colored lines represent the significance of apoptosis in HepG2 cells between each sample and its control. Controls are divided into (a) HepG2 cells (Black), (b) HepG2 cells transfected with pEGFP-N1 (Red), and (c) HepG2 cells transfected with pCIΔMEG3 (Blue). (**B**) Quantification of the stages of cell death after transfection with different plasmids. Data presented as mean ± SD. (*) shows the *p*-value < 0.05, (**) shows the *p*-value < 0.01 and (***) shows *p*-value < 0.001 compared to control untransfected HepG2 cells, pEGFP-N1 and pCIΔMEG3.

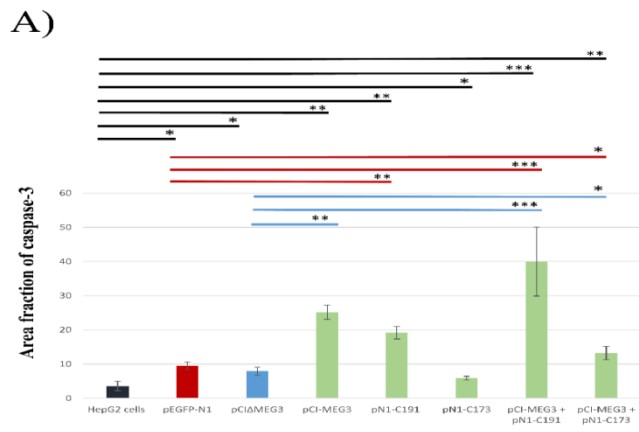

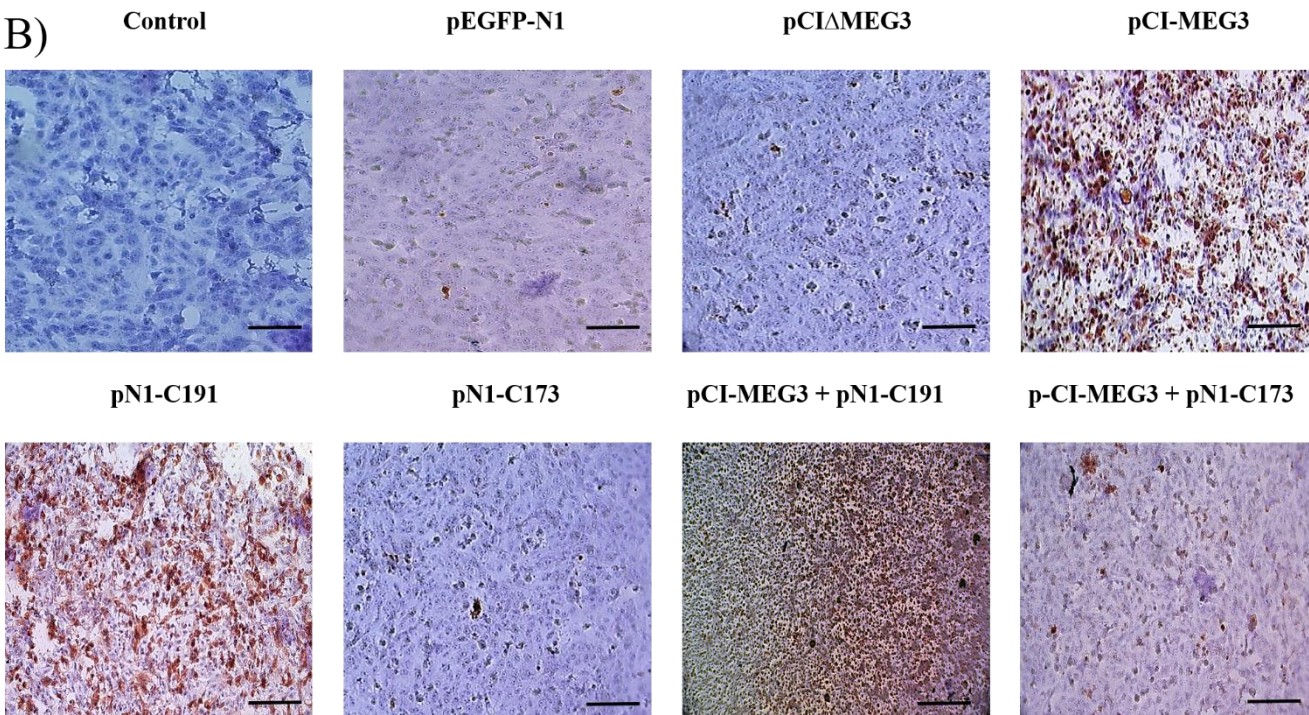

**Figure 3.** (**A**) Area fraction of caspase-3 expression level in HepG2 cells transfected with different plasmids. Data presented as mean ± SD. (*) shows the *p*-value < 0.05, (**) shows the *p*-value < 0.01 and (***) shows *p*-value < 0.001 compared to control untransfected HepG2 cells, pEGFP-N1 and pCIΔMEG3. Colored lines represent the significance of caspase-3 expression in HepG2 cells between each sample and its control. Controls are divided into (a) HepG2 cells (Black), (b) HepG2 cells transfected with pEGFP-N1 (Red), and (c) HepG2 cells transfected with pCIΔMEG3 (Blue). (**B**) Representative Photomicrographs of caspase-3 expression by ICC assay in transfected and co-transfected HepG2 cells with different plasmids. Scale bar 100 µm.

On the other hand, the results showed a significant decrease in the expression of MKI67 post-transfection with pCI-MEG3, pN1-C191, or pCI-MEG3 + pN1-C191. In contrast, there were no marked effects post-transfection with pN1-C173. However, co-transfection with pCI-MEG3 + pN1-C173 significantly increased the expression level of caspase-3 and decreased the expression level of MKI67 in HepG2 cells. (Figure 4A,B).

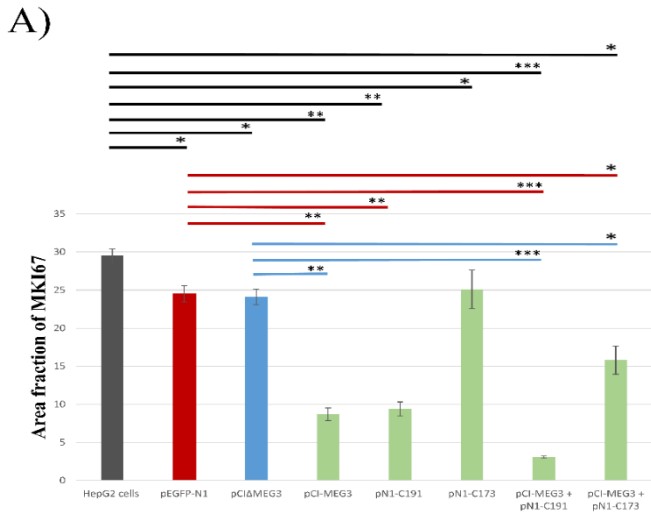

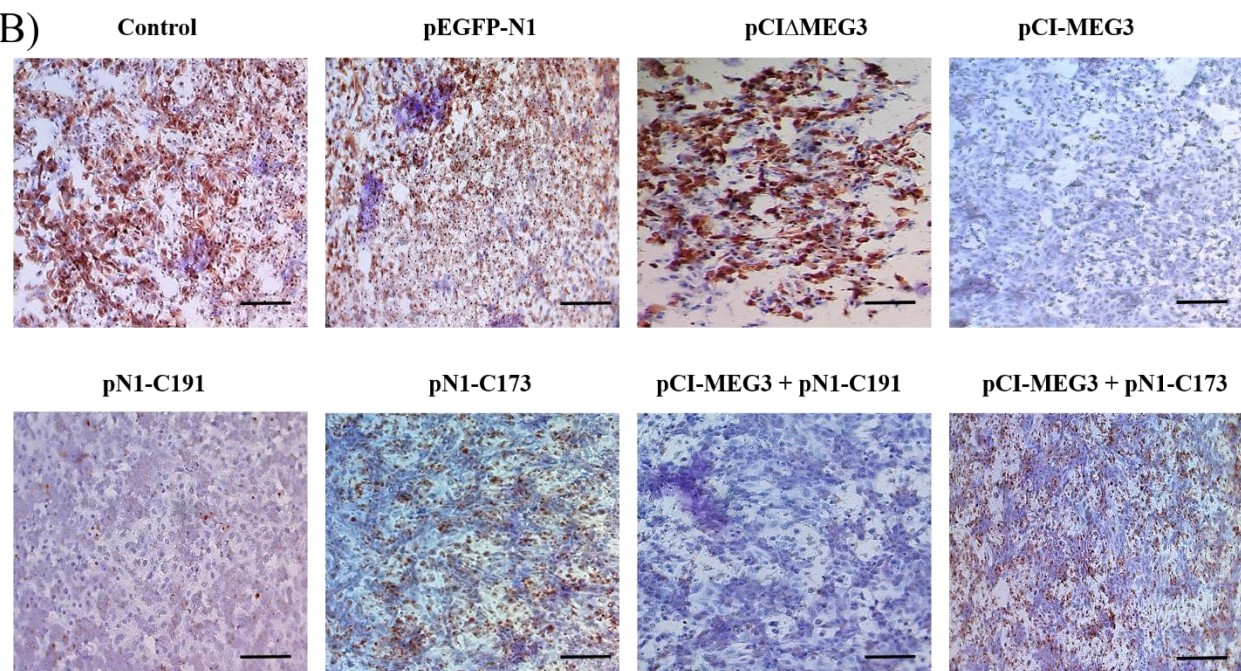

**Figure 4.** (**A**). Area fraction of MKI67 expression level in HepG2 cells transfected with different plasmids. Data presented as mean ± SD. (*) shows the *p*-value < 0.05, (**) shows the *p*-value < 0.01 and (***) shows *p*-value < 0.001 compared to control untransfected HepG2 cells, pEGFP-N1 and pCIΔMEG3. Colored lines represent the significance of MKI67 expression in HepG2 cells between each sample and its control. Controls are divided into (a) HepG2 cells (Black), (b) HepG2 cells transfected with pEGFP-N1 (Red), and (c) HepG2 cells transfected with pCIΔMEG3 (Blue). (**B**) Representative Photomicrographs of MKI67 expression by ICC assay in transfected and co-transfected HepG2 cells with different plasmids. Scale bar 100 μm.

*3.4. Co-Expression of lncRNA MEG3 and C191 Suppresses Proliferation of HepG2 Cells by Upregulating p53 and miRNA152 and Downregulation DNMT1, TGF-b, and BCL-2*

Changes in the transcripts levels of *p53*, *DNMT1*, *miRNA152*, *TGF-b*, and *BCL-2* target genes were validated by qRT-PCR post-transfection of HepG2 cells by pCI-MEG3, pN1-C191, pN1-173, pCI-MEG3 + pN1-C191, and pCI-MEG3 + pN1-C173. Transfection with pCI-MEG3 caused a significant increase in the transcripts levels of *p53* by around 5-fold

and *miRNA152* by around 6-fold while significantly decreasing the transcripts levels of *DNMT1*, *TGF-b*, and *BCL-2* by around 0.8-, 0.5-, and 0.65-fold, respectively. Furthermore, the results showed that transfection with pN1-C191 significantly increased the transcripts levels of *p53* and *miRNA152* by around 2.5-fold and 2.5-fold, respectively, while significantly decreasing the transcripts levels of *DNMT1*, *TGF-b*, and *BCL-2* by around 0.65-, 0.4-, and 0.4-fold, respectively. Meanwhile, co-transfection with pCI-MEG3 + pN1-C191 significantly increased the transcripts levels of *p53* and *miRNA152* by around 7-fold and 9-fold, respectively, while significantly decreasing the transcripts levels of *DNMT1*, *TGF-b*, and *BCL-2* by around 0.8-, 0.9-, and 0.85-fold, respectively. On the contrary, pN1-C173 significantly reduced the transcripts levels of *p53* and *miRNA152* by around 0.6- and 0.8-fold, respectively. Meanwhile, transfection with pN1-C173 significantly increased the transcript levels of *DNMT1*, *TGF-b*, and *BCL-2* by around 3-, 5-, and 4-fold, respectively. However, co-transfection with pCI-MEG3+ pN1-C173 significantly increased the transcripts levels of *p53* and *miRNA152* by around 2.5- and 3-fold, respectively, while significantly decreasing the transcripts levels of *DNMT1*, *TGF-b*, and *BCL-2* by around 0.5-, 0.2-, and 0.35-fold, respectively (Figure 5A–C).

A)

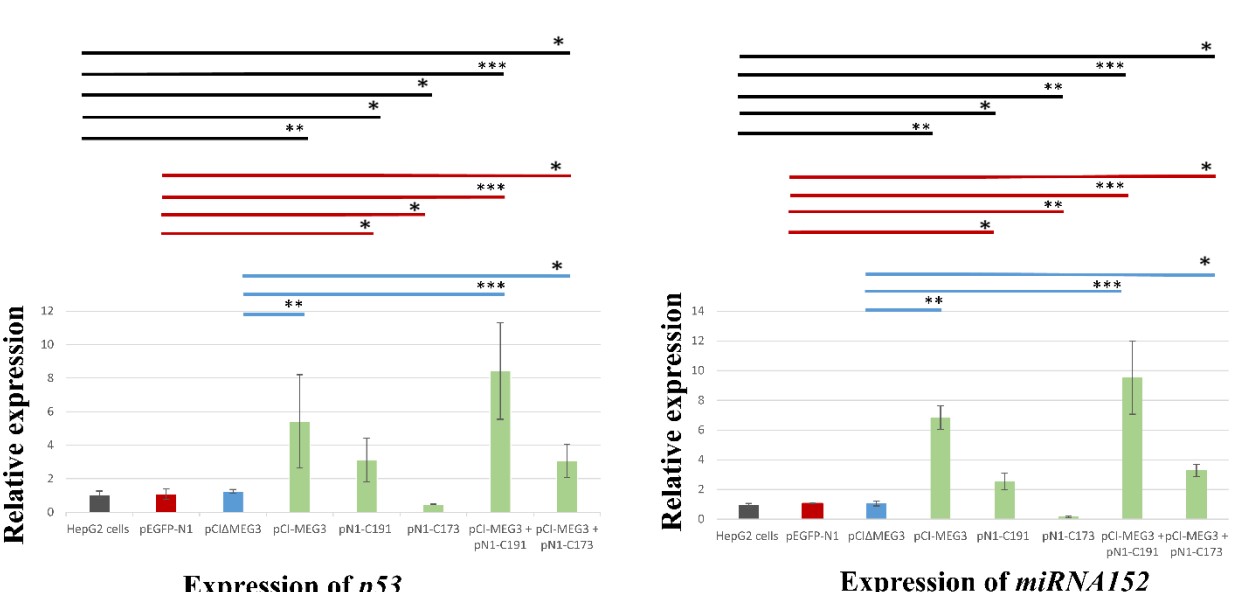

**Figure 5.** *Cont*.

B)

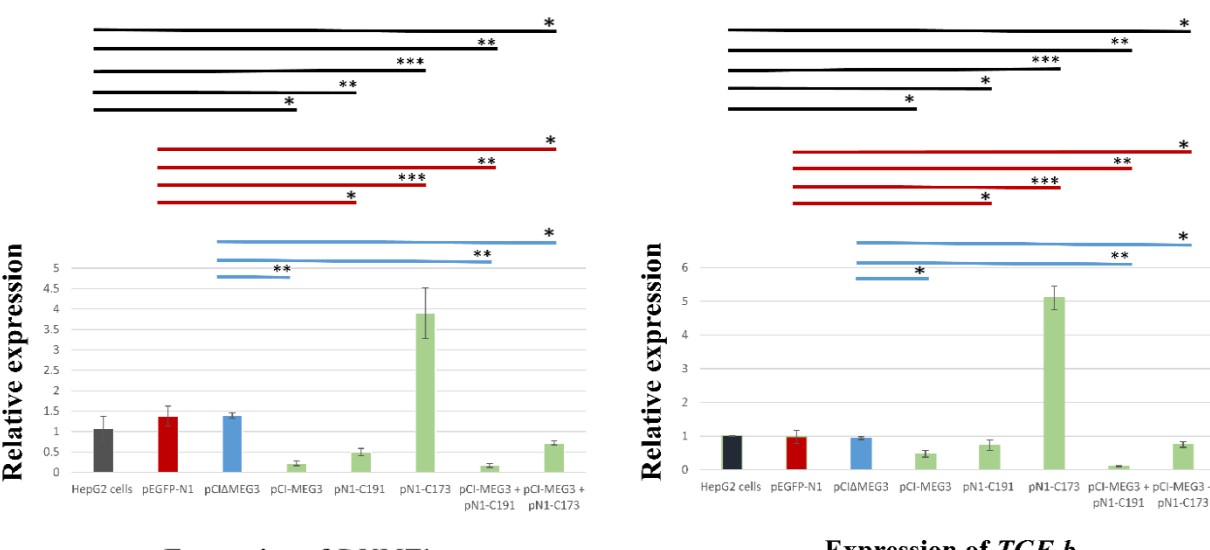

C)

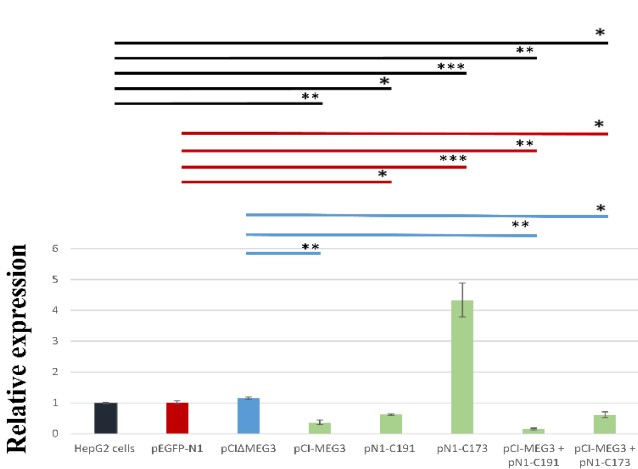

**Figure 5.** Quantification of the changes in expression of different cellular genes by qRT-PCR after transfection of HepG2 cells by different plasmids; pEGFP-N1, pCIΔMEG, pCI-MEG3, pN1-C191, pN1-C173, pCI-MEG3 + pN1-C191, and pCI-MEG3 + pN1-C173 compared to control untransfected HepG2 cells. The mean of three independent replicates was used to calculate these data values. Data presented as mean ± SD. (**A**) respresnts the relative expression change of P53 and miRNA152. (**B**) respresnts the relative expression change of DNMT1 and TGF-b. (**C**) respresnts the relative expression change of BCL-2. (*) represents *p*-value < 0.05, (**) represents *p*-value < 0.01, and (***) represents *p*-value < 0.001 compared to control untransfected HepG2 cell, pEGFP-N1, and pCIΔMEG3. Colored lines represent the significance of gene expression between each sample and its control. Controls are divided into (a) HepG2 cells (Black), (b) HepG2 cells transfected with pEGFP-N1 (Red), and (c) HepG2 cells transfected with pCIΔMEG3 (Blue).

## 4. Discussion

Many long non-coding RNAs (lncRNAs) are regulated in cancer cells; some of them are upregulated, and others are downregulated according to cancer types and compared to healthy cells [36]. In a different context, HCV core proteins are known to modulate many cellular pathways such as cell cycle, cell proliferation, apoptosis, and more. The dysregulation of these pathways by HCV core protein is thought to be involved in developing Hepatocellular Carcinoma (HCC) [4,5]. In this study, we were interested in evaluating the overexpression of one of the lncRNA (MEG3) that is known to be downregulated in HCC and determining the inverse capability of the overexpression of lncRNA MEG3 on HepG2 cells. Similarly, HCV core protein is also implicated in HCC development; the effect of the immature form (C191) and mature form (C173) of HCV core protein were evaluated alone or in combination with lncRNA MEG3 in HepG2 cells as a potential therapeutic approach. We found that the cell viability test of HepG2 cells after transfection with pCI-MEG3 revealed a marked difference between the effect of HCV core 191 and HCV core 173 on the cell viability, where cell viability of HepG2 cells was significantly decreased after transfection with pN1-C191 while it was not affected after transfection with pN1-C173. However, co-transfection of HepG2 cells with pCIMEG3 + pN1-C173 markedly decreased cell viability compared to transfection with pN1-C173 alone due to the effect of the lncRNA MEG3. This is supported by previous studies that revealed a significant decrease in cell viability by 19.3% and 30.8% after 48 h and 72 h, respectively, as a result of lncRNA MEG3 overexpression in HepG2 [37].

Cell morphology results supported the cell viability results as we observed an increase in accumulation of dead cells and signs of shrinkage and apoptotic bodies post-transfection with pCI-MEG3 and/or pN1-C191. On the other hand, there was no change in cell morphology after transfection with pN1-C173. The expression levels of five genes that are known to be regulated by lncRNA MEG3 and HCV core proteins were determined by qRT-PCR. A significant increase in the transcripts of *p53* and *miRNA152* and a significant decrease in *DNMT1, TGF-b,* and *BCL-2* were detected post-transfection with pCI-MEG3. However, there was a marked inverse effect between HCV core 191 (C191) and HCV core 173 (C173) on the transcripts levels of *p53, miRNA152, DNMT1, TGF-b,* and *BCL-2,* which reflects their effect on HepG2 cells proliferation. Transfection of HepG2 cells with pN1-C191 increased the transcripts levels of *p53* and *miRNA152* while significantly decreasing the transcripts levels of *DNMT1, TGF-b,* and *BCL-2.* This is similar to the results of the lncRNA MEG3 but with a lower effect. On the contrary, transfection of HepG2 cells with pN1-C173 has significantly decreased the transcripts levels of *p53* and *miRNA152* while significantly increasing the transcripts levels of *DNMT1, TGF-b,* and *BCL-2.* It is worth noting that the co-transfection of pCI-MEG3 and pN1-C191 showed more impact on transcription levels of these five genes than transfecting each vector alone; however, the main impact was due to lncRNA MEG3. Although HepG2 cells co-transfected with pCI-MEG3 and pN1-C173 overall showed the same transcription pattern as pCI-MEG3, the presence of C173 mitigated its effect. Our results are in accordance with a previous study, which reported that lncRNA MEG3 increases the transcriptional activity of the tumor suppressor p53 protein by interacting with its DNA binding domain (DBD), consequently influencing the expression of some of the genes regulated by p53 in hepatoma cells such as GADD45A, EGR1, SESN2 and TGFA [38]. A different study revealed that immature HCV core protein (C191) induces apoptosis in Huh7 cells by the upregulation of p53 via the suppression of oncoprotein casein kinase 1α (CK1α), a protein that has a role in tumor development and proliferation [39]. On the contrary, HCV core protein (C173) inhibits apoptosis and increases cell proliferation by different pathways; it can suppress the activity of tumor suppressor gene *p53* by interfering with promyelocytic leukemia isoform V (PML-IV), a major regulator of p53 function [40]. HCV core protein (C173) was shown to mediate HCC proliferation through downregulation of tumor suppressor miRNA 152 that suppresses oncoprotein wnt-1 [15], which has a vital role in developing the proliferation and progression of cancer cells [41,42]. It was proven that HCV core protein (C173) could downregulate tumor suppressor P16 expression

by upregulating DNA methyltransferase 1 (DNMT1), which increases hypermethylation of P16's promoter, leading to Rb inactivation and subsequent E2F1 activation and that leads to hepatocyte growth stimulation [16]. It is worth noting that the overexpression of DNMMT hyper-methylates the prompter of the lncRNA MEG3 gene, resulting in its downregulation in most cancerous cells [29]. Moreover, HCV core protein (173) induces HCC progression by inducing the upregulation of TGF-β1 transcription, where it binds at bases −376 to −331 bp in the promoter region of TGF-β1 and activates its expression [17]. On the contrary, LncRNA MEG3 was reported to inhibit cell growth and promote apoptosis via the TGF pathway [43].

In this study, there was a significant increase in total apoptosis (early and late) post-transfection with pCI-MEG3 and/or pN1-C191. Co-transfection with pCI-MEG3 and pN1-C173 showed a significant increase in total apoptosis (early and late), but transfection with only pN1-C173 did not. It is suggested that lncRNA MEG3 may mitigate the effect of HCV core 173 on HepG2 growth by increasing apoptosis. Our findings agree with a previous study by Zhang and his colleagues, who reported increased apoptosis of HepG2 cells overexpressing lncRNA MEG3 [44].

Finally, MKI67 expression is commonly used as a marker for tumor cell proliferation and growth [45]. The estimation of the expression levels of caspase-3 and MKI67 in HepG2 cells by immunocytochemistry assay showed a marked increase in caspase-3 and a marked decrease in MKI67 post-transfection with pCI-MEG3 and/or pN1-C191. On the contrary, C173 showed a marked decrease in caspase-3 and a marked increase in MKI67. The overexpression of LncRNA MEG3 with C173 mitigates the effect of C173 on cell proliferation. These results agree with a previous study, which found lncRNA MEG3 inhibits the development and proliferation of a laryngeal carcinoma xenograft tumor in nude mice [44]. The tumor growth was reduced in the lncRNA MEG3 group, as showed by a significant decrease in tumor weight and volume. In addition, tumor proliferation marker MKI67 was also reduced, and the expression of caspase-3 increased in the lncRNA MEG3 group [39].

In another study, the immunohistochemical results showed that HCV core (C173) increases the development and proliferation of HCC xenograft tumors in nude mice via the activation of the Wnt/β-catenin signaling pathway, which resulted in a marked increase in MKI67 compared to the control group [46].

## 5. Conclusions

Our results showed a marked upregulation of the tumor suppressor *p53* and *miRNA152* and downregulation of *DNMT1*, *BCL-2*, and *TGF-b* post-expression of lncRNA MEG3 with or without HCV core protein (C191). Additionally, an increase in apoptotic marker "caspase 3" and a decrease in tumor marker "MKI67" were detected after the expression of lncRNA MEG3 with or without HCV core protein (C191). On the contrary, results showed that HCV core protein (173) could mitigate the apoptotic effect of lncRNA MEG3 on HepG2 cells. Therefore, the expression of lncRNA MEG3 and HCV core protein (C191) induced apoptosis in the HepG2 cell line, whether they are expressed separately or in combination. This may suggest a potential anticancer characteristic against HCC, which should be confirmed by further studies.

**Author Contributions:** Literature search, D.M.; Lab work and methodology design, D.M. and T.Z.S.; data curation and analysis, D.M.; original draft preparation, D.M.; writing review and editing, T.Z.S.; data acquisition, T.Z.S. and S.S. visualization D.M.; supervision, T.Z.S., S.S. and A.A.B.; funding acquisition, T.Z.S. All authors have read and agreed to the published version of the manuscript.

**Funding:** This work is supported by the Science, Technology & Innovation Funding Authority (STDF) (grant No: 45396).

**Institutional Review Board Statement:** Not applicable.

**Informed Consent Statement:** Not applicable.

**Acknowledgments:** We wish to thank the Center for Aging and Associated Diseases (CAAD) and the center of the genome (CG), Zewail City of Science and Technology, for allowing us to use their facilities. Special thanks for Jihad I. Omran for his help and support in this work.

**Conflicts of Interest:** The authors declare no conflict of interest.

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
