# Peer review of "The Transgene Expression of the Immature Form of the HCV Core Protein (C191) and the LncRNA MEG3 Increases Apoptosis in HepG2 Cells"

_cimb, doi:10.3390/cimb44080249_

Round 1
Reviewer 1 Report
The paper is very interesting, the introduction is comprehensive and the methodology is conscientiously described. However, I have a few comments about the result section:
1. Microphotographs in the presented form are of very poor quality. In figure 3B, pN1-c191 seems differ in magnification. Ki67 is a nuclear protein but in figure 4B it's not so obvious (esp. CI-MEG3 + pN1-C173) – it's probably because of the poor quality of the microphotographs. Since they are an important part of this work they must be improved.
2. In my opinion the authors should be more careful in their conclusions. They have written about a „significant increase in total apoptosis” (in flow cytometry analysis). I understand that the increase in apoptotic cell population was statistically significant, but is a 2 -3% increase biologically relevant? An increase in caspase-3 expression, a decrease in BCL-2 expression is all that can be seen in these results. Of course, MTT test showed cytotoxicity, but we have to remember that it measures enzyme activity, not apoptosis.
3. Many researchers claim, and this is also my opinion, that we should use at least two reference genes in relative gene expression studies.
4. Information on repetitions is missing in the method section, Please add the n-value which was used in statistical calculations.
Author Response
Response to Reviewer 1 Comments
Comment 1: Microphotographs in the presented form are of very poor quality. In figure 3B, pN1 c191 seems differ in magnification. Ki67 is a nuclear protein but in figure 4B it's not so obvious (esp. CI-MEG3 + pN1-C173) – it's probably because of the poor quality of the microphotographs. Since they are an important part of this work they must be improved.
Response: Thank you for pointing this out. We agree with this comment. Therefore we have tried to improve the quality of microphotographs as shown in the revised copy of the paper with track changes. In addition, figures are provided in separate files for your convienence.
- For figure 3B, although the magnification was the same, however, we replaced the microphotograph of pN1-C191 with another better one.
- Regarding Ki67 microphotographs in figure 4B, we replaced treatment pCI-MEG3 and pN1-C173, with another microphotographs that clarified the difference in the expression level of Ki67 inside the nucleus.
Comment 2: In my opinion the authors should be more careful in their conclusions. They have written about a significant increase in total apoptosis” (in flow cytometry analysis). I understand that the increase in apoptotic cell population was statistically significant, but is a 2 -3% increase biologically relevant? An increase in caspase-3 expression, a decrease in BCL-2 expression is all that can be seen in these results. Of course, MTT test showed cytotoxicity, but we have to remember that it measures enzyme activity, not apoptosis.
Response: We measured apoptosis with different methods, which are a) flowcytometry through measuring the level of phosphatidylserine (PS) with Annexin V-FITC Apoptosis kit, b) immunocytochemistry against cleaved Caspase 3, c) qRT-PCR for BCL-2. However, we removed the word ‘significant’ from the part of the results of measurement of the apoptosis by flowcytometry to be
“Flow cytometry was used to determine the fraction of cells undergoing apoptosis (Figure 2). The results showed an increase of early and late apoptosis by 5.2 and 3.2 folds, respectively, post co-transfection of HepG2 cells with pCI-MEG3+ pN1-C191 compared to the control (untransfected HepG2 cells) (Figure 2A). This increase in early and late apoptosis was maintained when compared to HepG2 cells transfected with pEGFP-N1 or pCI∆MEG3 by approximately 4.0 and 2.5 folds, respectively. Additionally, there was an increase in early and late apoptosis after transfection with pCI-MEG3 by 3.2 and 3.0 folds, respectively, compared to untransfected HepG2 cells and by 2.5 and 2.3 folds, respectively, compared to cells transfected with pCI∆MEG3. pN1-C191 showed an increase in early apoptosis by 3.5 folds, but no increase in late apoptosis was detected. Co-transfection with pCI-MEG3 + pN1-C173 increased early and late apoptosis by 2.6 and 2.4 folds, respectively, compared to control untransfected HepG2 cells, and by approximately 2 and 1.7 folds, respectively, compared to pEGFP-N1 or pCI∆MEG3. Transfection with pN1-C173 did not cause an effect on early or late apoptosis (Figure 2 A and B).”
Instead of
“Flow cytometry was used to determine the fraction of cells undergoing apoptosis (Figure 2). The results showed a significant increase of early and late apoptosis by 5.2 and 3.2 folds, respectively, post co-transfection of HepG2 cells with pCI-MEG3+ pN1-C191 compared to the control (untransfected HepG2 cells) (Figure 2A). This significant increase of early and late apoptosis was maintained when compared to HepG2 cells transfected with pEGFP-N1 or pCI∆MEG3 by approximately 4.0 and 2.5 folds, respectively. Additionally, there was a significant increase in early and late apoptosis after transfection with pCI-MEG3 by 3.2 and 3.0 folds, respectively, compared to untransfected HepG2 cells and by 2.5 and 2.3 folds, respectively, compared to cells transfected with pCI∆MEG3. pN1-C191 showed a significant increase in early apoptosis by 3.5 folds, but no increase in late apoptosis was detected. Co-transfection with pCI-MEG3 + pN1-C173 significantly increased early and late apoptosis by 2.6 and 2.4 folds, respectively, compared to control untransfected HepG2 cells, and by approximately 2 and 1.7 folds, respectively, compared to pEGFP-N1 or pCI∆MEG3. Transfection with pN1-C173 did not cause an effect on early or late apoptosis (Figure 2 A and B).”
Comment 3: Many researchers claim, and this is also my opinion, that we should use at least two reference genes in relative gene expression studies.
Response: Thanks a lot for your valuable comment, it would have been interesting to explore this aspect. However, in the case of our study, using another reference gene will require repeating all the experiments. Also, the qRT-PCR results are in agreement with the other experiments and does not contradict them. In addition, many works adopt using one reference including our previous work https://www.sciencedirect.com/science/article/abs/pii/S0378111921001359?via%3Dihub
https://www.ncbi.nlm.nih.gov/pmc/articles/PMC6753855/
Comment 4: Information on repetitions is missing in the method section, Please add the n value which was used in statistical calculations.
Response: We agree with this and have incorporated the suggested comment, therefore the information of repetitions is added in the section of methodology, part of the statical analysis.
Reviewer 2 Report
Major concern- The constructs used in modulating cell viability of Hep-G2 cells appear to have varying levels of impact. I assume you have checked for the efficiency of transformation of your constructs(?) but also, of more concern, have you checked for the level of expression of each of the constructs not only at 48hr and 72 hr (Fig1), and also in the subsequent experiments of apoptosis and downstream targets? This check for the presence of the construct may help explain variability and may also why there is no additive affect of MEG3 and C191. This is critical and without this information, the significance of subsequent experiments is lost as you have not shown a correlation between construct expression and affect.
Minor concern- There is no explanation in the legends as to what the (Black, blue, red) horizontal lines above each of the Figures represents.
Round 2
Reviewer 1 Report
The photos are still of poor quality - I think there is some problem with creating figures from single photos. Moreover, the authors replied that they had included files with single photographs, but I do not see them in the available materials.
Author Response
Comments from Reviewer 1
The photos are still of poor quality - I think there is some problem with creating figures from single photos. Moreover, the authors replied that they had included files with single photographs, but I do not see them in the available materials.
Response : Thank you for your comments and your recommendation, we uploaded the photos with better resolution. We indeed uploaded the photos separately along with the ones included in the revised manuscript and we included them with the response to reviewers again this time for your convenience.

Reviewer 2 Report
In the revised manuscript, the table of oligonucleotide primers has been submitted by the authors to support the expression of constructs introduced into the HepG2 cells and to measure the expression of potential targets p53, miR-152, Tgfb, Bcl2 and DNMT. There are still problems with the data as presented, stemming primarily from the fact that several of the oligonucleotide primers do not appear to match the genes as intended. Specifically, no significant match has been found for the MEG3 5' primer, Tgfb 5' primer, or DNMT 5' primer. Furthermore, while there are matches found for the 3' primers for MEG3 and DNMT, the lack of recognition that there are multiple transcript variants for both DNMT and MEG3, removes any confidence in the data presented in section 3.4 that targets are up-regulated or down-regulated.
Author Response
Comments from Reviewer 2
In the revised manuscript, the table of oligonucleotide primers has been submitted by the authors to support the expression of constructs introduced into the HepG2 cells and to measure the expression of potential targets p53, miR-152, Tgfb, Bcl2 and DNMT. There are still problems with the data as presented, stemming primarily from the fact that several of the oligonucleotide primers do not appear to match the genes as intended. Specifically, no significant match has been found for the MEG3 5' primer, Tgfb 5' primer, or DNMT 5' primer. Furthermore, while there are matches found for the 3' primers for MEG3 and DNMT, the lack of recognition that there are multiple transcript variants for both DNMT and MEG3, removes any confidence in the data presented in section 3.4 that targets are up-regulated or down-regulated.
Response : Thanks a lot for your comments, however we are confused about the comment that several of the oligonucleotide primers do not appear to match the genes as intended. We checked all the primers again and their sequences are similar to primers that already published before. Below are the response of each of them and the source of the papers that match their sequences:
- For MEG3 primers ( Forward and reverse): primers were designed according to the MEG3 gene that we used to transfect cells by plasmid pCI-MEG3, which was purchased from Addgene (Catalog no:44727). We encourage the reviewer to map the MEG3 primers to the pCI-MEG3 and to check the paper cited from Addgene for pCI-MEG3
https://www.jbc.org/article/S0021-9258(18)80958-9/fulltext
- For p53 primers : we rechecked the primers again and they matched the p53 sequance.
In addition, the same primers sequences are listed in these already published papers
https://www.ncbi.nlm.nih.gov/pmc/articles/PMC6042264/table/Taba/
https://www.hindawi.com/journals/sci/2019/6804036/tab1/
- For DNMT1, we rechecked the primers again and they matched the DNMT1 sequance.
In addition, the same primers sequences are listed in these already published papers
https://www.ncbi.nlm.nih.gov/pmc/articles/PMC4905493/
https://www.liebertpub.com/doi/pdf/10.1089/gtmb.2018.0285
- For miRNA152, the primer was listed in this already published paper.
https://www.ncbi.nlm.nih.gov/pmc/articles/PMC8809980/
- For BCL2: same primers sequences are listed in these already published papers
https://www.nature.com/articles/cr2016159
https://www.hindawi.com/journals/omcl/2019/7824684/tab1/
- For TGFb: the primers are matched with a previous paper
https://www.spandidos-publications.com/10.3892/ol.2019.11189
- We also confused about the reviewer’s comment about DNMT and MEG3 variants, as we mentioned that we measure the expression of DNMT1 specifically not anyother variants such as DNMT3A or 3B. In addition, we used primers specific for the MEG3 in the addgene vector and this is what we measured to prove its expression in the examined cells post transfection.
Round 3
Reviewer 2 Report
The MEG3 primer issue is two-fold: i). where do the MEG3 forward and reverse primers map with respect to the MEG3 lncRNA and ii). alternatively spliced variant MEG 3a, 3b, 3c and 3d. In an oligo blast search using the forward primer for MEG3, no matches are found. Downloading the paper that the authors cite (Zhou et al., 2007) that describes the cloning and structure of the MEG3 splice variants (and the primers used by the authors for MEG3, no match is found in a search of the entire sequence (NR_002766 in GENBANK). It's possible that the primer sequence is accurate and it is somewhere in that sequence (I wouldn't rely on Addgene for the positioning), but what would be helpful is some description from the authors as to where the MEG3 primers used in the study actually map (in terms of splice variants, or which exon, some clue as to where one can find the primer), so that a reader could reproduce the study if desired.
The same question applies to the DMNT1 variants, where do the primers map with regard to the variant sequences so the reader can determine if what DMNT1 variants are being targeted in the experiments.
Author Response
Comments from Reviewer 2
The MEG3 primer issue is two-fold: i). where do the MEG3 forward and reverse primers map with respect to the MEG3 lncRNA and ii). Alternatively spliced variant MEG 3a, 3b, 3c and 3d. In an oligo blast search using the forward primer for MEG3, no matches are found. Downloading the paper that the authors cite (Zhou et al., 2007) that describes the cloning and structure of the MEG3 splice variants (and the primers used by the authors for MEG3, no match is found in a search of the entire sequence (NR_002766 in GENBANK). It's possible that the primer sequence is accurate and it is somewhere in that sequence (I wouldn't rely on Addgene for the positioning), but what would be helpful is some description from the authors as to where the MEG3 primers used in the study actually map (in terms of splice variants, or which exon, some clue as to where one can find the primer), so that a reader could reproduce the study if desired.
The same question applies to the DMNT1 variants, where do the primers map with regard to the variant sequences so the reader can determine if what DMNT1 variants are being targeted in the experiments.
Response:
We would like to thank you for your valuable comments and recommendation, which will add to our paper. As previously indicated the forward primer of the MEG3 was designed based on the sequence of the Addgene vector pCI-MEG3 # 44727. The forward primer is located downstream of the chimeric intron and upstream of the MEG3 gene. We used the forward primer to target the 5’end of the transcript and the reverse primer to target the MEG3 sequence itself. The main purpose of that is to detect specifically the expression of the MEG3 from the transfected pCI-MEG3 and not the endogenous MEG3 lncRNA to confirm the successful transfection of pCI-MEG3. The manuscript was modified to include this information in pages 233-235
Kindly find below the position of forward primer on the pCI-MEG3 (https://www.addgene.org/browse/sequence/61283/). It is worth noting the MEG3 cDNA was cloned into pCI mammalian expression vector https://pl.promega.com/-/media/files/resources/protocols/technical-bulletins/0/pci-mammalian-expression-vector-protocol.pdf?sc_lang=en
> pCI-MEG3 sequence 5613 bps
TCAATATTGGCCATTAGCCATATTATTCATTGGTTATATAGCATAAATCAATATTGGCTATTGGCCATTG
CATACGTTGTATCTATATCATAATATGTACATTTATATTGGCTCATGTCCAATATGACCGCCATGTTGGC
ATTGATTATTGACTAGTTATTAATAGTAATCAATTACGGGGTCATTAGTTCATAGCCCATATATGGAGTT
CCGCGTTACATAACTTACGGTAAATGGCCCGCCTGGCTGACCGCCCAACGACCCCCGCCCATTGACGTCA
ATAATGACGTATGTTCCCATAGTAACGCCAATAGGGACTTTCCATTGACGTCAATGGGTGGAGTATTTAC
GGTAAACTGCCCACTTGGCAGTACATCAAGTGTATCATATGCCAAGTCCGCCCCCTATTGACGTCAATGA
CGGTAAATGGCCCGCCTGGCATTATGCCCAGTACATGACCTTACGGGACTTTCCTACTTGGCAGTACATC
TACGTATTAGTCATCGCTATTACCATGGTGATGCGGTTTTGGCAGTACACCAATGGGCGTGGATAGCGGT
TTGACTCACGGGGATTTCCAAGTCTCCACCCCATTGACGTCAATGGGAGTTTGTTTTGGCACCAAAATCA
ACGGGACTTTCCAAAATGTCGTAATAACCCCGCCCCGTTGACGCAAATGGGCGGTAGGCGTGTACGGTGG
GAGGTCTATATAAGCAGAGCTCGTTTAGTGAACCGTCAGATCACTAGAAGCTTTATTGCGGTAGTTTATC
ACAGTTAAATTGCTAACGCAGTCAGTGCTTCTGACACAACAGTCTCGAACTTAAGCTGCAGAAGTTGGTC
GTGAGGCACTGGGCAGGTAAGTATCAAGGTTACAAGACAGGTTTAAGGAGACCAATAGAAACTGGGCTTG
TCGAGACAGAGAAGACTCTTGCGTTTCTGATAGGCACCTATTGGTCTTACTGACATCCACTTTGCCTTTC
TCTCCACAGGTGTCCACTCCCAGTTCAATTACAGCTCTTAAGGCTAGAGTACTTAATACGACTCACTATA
GGCTAGCCTCGAGAATTCGCAGAGAGGGAGCGCGCCTTGGCTCGCTGGCCTTGGCGGCGGCTCCTCAGGA
GAGCTGGGGCGCCCACGAGAGGATCCCTCACCCGGGTCTCTCCTCAGGGATGACATCATCCGTCCACCTC
CTTGTCTTCAAGGACCACCTCCTCTCCATGCTGAGCTGCTGCCAAGGGGCCTGCTGCCCATCTACACCTC
ACGAGGGCACTAGGAGCACGGTTTCCTGGATCCCACCAACATACAAAGCAGCCACTCACTGACCCCCAGG
ACCAGGATGGCAAAGGATGAAGAGGACCGGAACTGACCAGCCAGCTGTCCCTCTTACCTAAAGACTTAAA
CCAATGCCCTAGTGAGGGGGCATTGGGCATTAAGCCCTGACCTTTGCTATGCTCATACTTTGACTCTATG
AGTACTTTCCTATAAGTCTTTGCTTGTGTTCACCTGCTAGCAAACTGGAGTGTTTCCCTCCCCAAGGGGG
TGTCAGTCTTTGTCGACTGACTCTGTCATCACCCTTATGATGTCCTGAATGGAAGGATCCCTTTGGGAAA
TTCTCAGGAGGGGGACCTGGGCCAAGGGCTTGGCCAGCATCCTGCTGGCAACTCCAAGGCCCTGGGTGGG
CTTCTGGAATGAGCATGCTACTGAATCACCAAAGGCACGCCCGACCTCTCTGAAGATCTTCCTATCCTTT
TCTGGGGGAATGGGGTCGATGAGAGCAACCTCCTAGGGTTGTTGTGAGAATTAAATGAGATAAAAGAGGC
CTCAGGCAGGATCTGGCATAGAGGAGGTGATCAGCAAATGTTTGTTGAAAAGGTTTGACAGGTCAGTCCC
TTCCCACCCCTCTTGCTTGTCTTACTTGTCTTATTTATTCTCCAACAGCACTCCAGGCAGCCCTTGTCCA
CGGGCTCTCCTTGCATCAGCCAAGCTTCTTGAAAGGCCTGTCTACACTTGCTGTCTTCCTTCCTCACCTC
CAATTTCCTCTTCAACCCACTGCTTCCTGACTCGCTCTACTCCGTGGAAGCACGCTCACAAAGGCACGTG
GGCCGTGGCCCGGCTGGGTCGGCTGAAGAACTGCGGATGGAAGCTGCGGAAGAGGCCCTGATGGGGCCCA
CCATCCCGGACCCAAGTCTTCTTCCTGGCGGGCCTCTCGTCTCCTTCCTGGTTTGGGCGGAAGCCATCAC
CTGGATGCCTACGTGGGAAGGGACCTCGAATGTGGGACCCCAGCCCCTCTCCAGCTCGAAATCCCTCCAC
AGCCACGGGGACACCCTGCACCTATTCCCACGGGACAGGCTGGACCCAGAGACTCTGGACCCGGGGCCTC
CCCTTGAGTAGAGACCCGCCCTCTGACTGATGGACGCCGCTGACCTGGGGTCAGACCCGTGGGCTGGACC
CCTGCCCACCCCGCAGGAACCCTGAGGCCTAGGGGAGCTGTTGAGCCTTCAGTGTCTGCATGTGGGAAGT
GGGCTCCTTCACCTACCTCACAGGGCTGTTGTGAGGGGCGCTGTGATGCGGTTCCAAAGCACAGGGCTTG
GCGCACCCCACTGTGCTCTCAATAAATGTGTTTCCTGTCTTAACAAAAAAAAAAAAAAAAAAAAAAAAAA
AAAAAAAAAAAAAAAAAAAAAAAAAAAAAAAAAAAAAAAAAAAGCGGCCGCTTCGAGCAGACATGATAAG
For DNMT1: The primers are specific for different variants of Homo sapiens DNMT1 on NCBI:
- Homo sapiens DNA methyltransferase 1 (DNMT1), transcript variant 2, mRNA
NCBI Reference Sequence: NM_001379.4.
- Homo sapiens DNA methyltransferase 1 (DNMT1), transcript variant 4, mRNA
NCBI Reference Sequence: NM_001318731.2
3- Homo sapiens DNA methyltransferase 1 (DNMT1), transcript variant 3, mRNA
NCBI Reference Sequence: NM_001318730.2

Round 4
Reviewer 2 Report
The inclusion of the information that the 5' forward MEG3 primer is actually the CI vector based and not MEG3 alone is very critical information that clarifies the results and findings. Much better!!
As for the DNMT primers recognizing some of the isoforms of DNMT1, there should be some reference to this fact in the paper, probably in the M&M (quantitative real time PCR section) or in the Results lines 501-512 where the DNMT1 transcripts are discussed.
Author Response
Comments from Reviewer 2
The inclusion of the information that the 5' forward MEG3 primer is actually the CI vector based and not MEG3 alone is very critical information that clarifies the results and findings. Much better!!
As for the DNMT primers recognizing some of the isoforms of DNMT1, there should be some reference to this fact in the paper, probably in the M&M (quantitative real time PCR section) or in the Results lines 501-512 where the DNMT1 transcripts are discussed.
Response: Thank you again for your valuable comments; we added the details of MEG3 primer design in the previous uploaded version of the manuscript. Also, we have added in the section of M&M, page 5, line 235 this sentence “All the primers were designed to match most of the transcript variants of the tested genes” to clarify the fact that all genes’ primers were designed to cover most of the genes’ variants.